# Subpixel Matching Using Double-Precision Gradient-Based Method for Digital Image Correlation

**DOI:** 10.3390/s21093140

**Published:** 2021-04-30

**Authors:** Gang Liu, Mengzhu Li, Weiqing Zhang, Jiawei Gu

**Affiliations:** School of Civil Engineering, Chongqing University, No. 83 Shabei Street, Chongqing 400045, China; 20161602020@cqu.edu.cn (M.L.); zhangweiqing@tyut.edu.cn (W.Z.); 201916131169@cqu.edu.cn (J.G.)

**Keywords:** displacement measurement, digital image correlation, subpixel matching, gradient-based algorithm, linear combination

## Abstract

Digital image correlation (DIC) for displacement and strain measurement has flourished in recent years. There are integer pixel and subpixel matching steps to extract displacement from a series of images in the DIC approach, and identification accuracy mainly depends on the latter step. A subpixel displacement matching method, named the double-precision gradient-based algorithm (DPG), is proposed in this study. After, the integer pixel displacement is identified using the coarse-fine search algorithm. In order to improve the accuracy and anti-noise capability in the subpixel extraction step, the traditional gradient-based method is used to analyze the data on the speckle patterns using the computer, and the influence of noise is considered. These two nearest integer pixels in one direction are both utilized as an interpolation center. Then, two subpixel displacements are extracted by the five-point bicubic spline interpolation algorithm using these two interpolation centers. A novel combination coefficient considering contaminated noises is presented to merge these two subpixel displacements to obtain the final identification displacement. Results from a simulated speckle pattern and a painted beam bending test show that the accuracy of the proposed method can be improved by four times that of the traditional gradient-based method that reaches the same high accuracy as the Newton–Raphson method. The accuracy of the proposed method efficiently reaches at 92.67%, higher than the Newton-Raphon method, and it has better anti-noise performance and stability.

## 1. Introduction

Using displacement, strain, and vibration-based measurements, structural damage detection and integrity assessment have become popular in the last two decades [1]. The displacement response of a structure exhibits a potential to more accurately detect damage since it directly reflects the structural overall stiffness [2]. However, displacement measurement is very difficult to apply for realistic engineering structures due to the expensive cost such as the global navigation satellite system, or incapable requirements such as stationary reference points. Fortunately, a camera-based vision measurement, namely digital image correlation (DIC), has emerged in recent years as a promising alternative to these aforementioned displacement measurement technologies [3]. This approach can provide a relatively continuous measurement on the patterned area of a structure and can be called a full-field measurement technique, essentially within the entire line of sight of the cameras, and DIC has been widely used in different fields [4,5].

Digital image correlation (DIC) is an optical measurement technique that works based on gray-scale variations of continuous patterns. There are two main steps to implement DIC for displacement extraction: integral pixel and subpixel level matching. A multitude of integral pixel algorithms, such as the coarse-fine search and diamond search, have been proposed [6], and the performance of these methods has been satisfactorily verified by tests [7]. Chen et al. proposed an adaptive point cloud correction algorithm designed to optimize the point cloud structure after image stitching of a multi-camera and improve the accuracy of surface reconstruction [8]. Tang et al. applied the algorithm to the dynamic real-time detection of surface deformation and full-field strain in recycled aggregate concrete-filled steel tubular columns [9,10]. Rizo-Patron used DIC to accurately identify the modal parameters of rotating helicopter rotor blades [11]. Since digital images record discrete grayscale information, no matter how the defined correlation function is utilized for the correlation search, the translation of the subset can only be carried out in the unit of integer pixel; therefore, displacement resolution is limited and cannot meet the requirement for detecting damage in many cases.

Subpixel displacement extraction, which is performed based on the results of the integral pixel level identification, is considered a key technique to improve DIC-based measurement resolution. A sea of subpixel matching algorithms based on correlation coefficients has been presented, such as curved fitting [12], cross search [13], Newton–Raphson (N–R) [14], and gradient-base (GB) methods [15]. Among these approaches, the frequently utilized algorithms are the N–R and GB methods [16]. The N–R algorithm implements an iterative procedure through optimizing a non-linear correlation function to acquire accurate subpixel extraction. Nevertheless, one inherent limitation or computational burden is that the Hessian matrix must be re-evaluated and inverted in every iterative procedure [17,18]. The GB method utilizes the gray gradient or the statistical properties of micro-regions to obtain the subpixel displacement [19,20], hence, the calculation efficiency is superior to the N–R method. The accuracy of the GB method, however, is lower than the N–R algorithm [21].

Although the GB method has the advantage of fast calculation, its accuracy gradually decreases with the increase of image noise. This paper is motivated by this feature of the GB method to try to find a subpixel algorithm that can inherit the high efficiency of the GB method while having the characteristics of high precision. Furthermore, the accuracy of the GB method is greatly affected by the gray gradient, and the gray gradient is usually acquired from the bicubic spline interpolation algorithm using five-point pixels, whose accuracy is determined by the center selection of interpolation. While feasible points as the interpolation center have been given in the integral pixel matching step, there is still a paradox of which point is the optimal center. To address this problem, the double precision gradient-based (DPG) method, which is a subpixel matching algorithm with high efficiency and precision, is proposed in this study. Two interpolation functions that take the nearest neighborhood integer points as the center of the interpolation are implemented to acquire the corresponding subpixel displacement. A weight coefficient is introduced and then the identified displacement is linearly combined to obtain the final subpixel matching result. One advantage of the proposed DPG method is that it not only maintains high computational efficiency but also improves the accuracy of the conventional GB method. Furthermore, the DPG method shows a good anti-noise capability because the influence of noise is considered during interpolation. 

This paper starts with a brief description of traditional integral pixel and subpixel matching methods in Section 2. The proposed DPG method for subpixel matching is then presented in Section 3. In Section 4, a numerical simulation and an experimental test are conducted to verify the proposed method. Finally, the conclusions are given in Section 5.

## 2. Matching Algorithm for DIC

### 2.1. Integral Pixel Level Matching

The basic principle of the standard GB method is to track the same subimage (or subset) located in the reference image and deformed images to retrieve the full-field displacement. Typically, a unified Cadier coordinate system is established for the reference and deformed images as shown in Figure 1. For a pixel in the reference image, marked as the point *P*, a subset with (2*M* + 1) × (2*M* + 1) pixels centered at point *P* is chosen, where *M* is radius of the subset. Correspondingly, a stochastic point is picked as the initial search point *P*′ in the deformed image, and a target subset with (2*M* + 1) × (2*M* + 1) pixels centered at *P*′ is selected as the target subset. A pre-defined correlation ZNSSD criterion is then performed to assess the similarity between these two subsets. For its good accuracy and anti-noise capability, the standardized covariance cross-correlation coefficient *C*(•) is mostly used and can be defined as [22],
(1)$$C(u,v)=\sum_{x=-M}^{M}\sum_{y=-M}^{M}{\left[\frac{f(x,y)-f_{m}}{\sqrt{\sum_{x=-M}^{M}\sum_{y=-M}^{M}{\left[f(x,y)-f_{m}\right]}^{2}}}-\frac{g(x+u,y+v)-g_{m}}{\sqrt{\sum_{x=-M}^{M}\sum_{y=-M}^{M}{\left[g(x+u,y+v)-g_{m}\right]}^{2}}}\right]}^{2}$$
where *x*,*y* are position of the point *P* along *x*-*y* direction in the coordinate system, respectively; *u* and *v* are displacement between point *P* and *P*′ along *x*-*y* direction, respectively; *f*(*x*, *y*) and *g*(*x* + *u*, *y* + *v*) are the gray values of the subset in the reference and deformed image, respectively; and *g_m_* and *f_m_* are the ensemble averages of the selected subset in reference and target subset, respectively.

The (*u*,*v*) with maximum *C*(*u*,*v*) is considered the true displacement for point *P*. Thus, the process of integral pixel level matching is actually to search an optimal matching subset in the deformed image. Several integral pixel search algorithms are proposed to determine the parameter M and the search procedure for efficient matching, such as coarse-fine, three steps, and diamond search methods [23]. The coarse-fine algorithm was adopted in this paper because it is the most commonly used integral pixel search algorithm and has been successfully verified in many fields [24]. The procedure of this algorithm includes: firstly, according to the pre-estimated displacement, determine the search area with a random *P*’ point in the deformed image as the center, and use Equation (1) to find the translation solution *u_pt_* and *v_pt_* with the global maximum correlation coefficient in the search area. Secondly, take *u_pt_* and *v_pt_* t as the initial value, assuming that the shear deformation parameters ∂*u_pt_*/∂*y* and ∂*v_pt_*/∂*x* are zero, estimate the tensile or compression deformation parameters ∂*u_pt_*/∂*x* and ∂*v_pt_*/∂*y*, and use the same search strategy as the first step until the optimal solution ∂*u_pt_*/∂*x* and ∂*v_pt_*/∂*y* are found. Finally, use the *u_pt_*, *v_pt_*, ∂*u_pt_*/∂*x* and ∂*v_pt_*/∂*y* obtained in the first two steps as initial values, use the previous search method to find the optimal solutions ∂*u_pt_*/∂*y* and ∂*v_pt_*/∂*x*. The first-order mapping function is used to combine *u_pt_*, *v_pt_*, ∂*u_pt_*/∂*x*, ∂*v_pt_*/∂*y*, ∂*u_pt_*/∂*y*, and ∂*v_pt_*/∂*x* to get the integral pixels *u_p_* and *v_p_*. The detailed procedure of the coarse-fine algorithm can be found in reference [23].

It should be noted that the minimal unit in a digital image is one pixel; the displacement identified from integral pixel level matching is an integer multiple of one pixel, that is, the acquired *u* and *v* are integer numbers. To further improve the registration resolution, the subpixel matching algorithm has to be implemented.

### 2.2. Subpixel Level Matching Based on Gradient-Based (GB) Method

The gradient-based (GB) method was first developed by Davis and Freeman as an optical flow method [25]. Based on the basic assumption of the GB method, the subset rigid body translation exists when the subset is small enough, that is,
(2)f(x,y)=g(x+u+Δx,y+v+Δy)
where Δ*x* and Δ*y* are the subpixel displacements along the *x*-*y* direction, respectively. After neglecting the high-order terms, the first-order Taylor’s expansion of function *g*(•) yields
(3)$$\begin{array}{l} g(x+u+\Delta x,y+v+\Delta y)\\ =g(x+u,y+v)+\Delta x\cdot g_{x}(x+u,y+v)+\Delta y\cdot g_{y}(x+u,y+v) \end{array}$$
where *g_x_*, *g_y_* are the first-order derivatives of intensities at the center of the best matching subset in the target image. According to reference [26], Equation (3) can be solved using the least squares technique with the following closed form:(4)$$\left[\begin{array}{c} \Delta x\\ \Delta y \end{array}\right]={\left[\begin{array}{cc} \sum \sum {\left(g_{x}\right)}^{2} & \sum \sum \left(g_{x}\cdot g_{y}\right)\\ \sum \sum \left(g_{x}\cdot g_{y}\right) & \sum \sum {\left(g_{y}\right)}^{2} \end{array}\right]}^{-1}\cdot \left[\begin{array}{c} \sum \sum \left[(f-g)g_{x}\right]\\ \sum \sum \left[(f-g)g_{y}\right] \end{array}\right]$$
where *g_x_*(*x* + *u*,*y* + *v*) and *g_y_*(*x* + *u*,*y* + *v*) are abbreviated as *g_x_* and *g_y_*, respectively for simple expressions. *f*(*x*,*y*) and *g*(*x* + *u*,*y* + *v*) are substituted by simple description *f* and *g*, respectively. Equation (4) indicates that the key step to acquire the subpixel displacement depends on the gradient operators *g_x_* and *g_y_*.

Since the value of grayscale intensity function *g*(•) is only available at the integer pixel point, the interpolation scheme is usually adopted to acquire gradient value for *g_x_* and *g_y_*. The Not-a-Knot interpolation algorithm, one of the bicubic spline interpolation algorithms using five-point pixels, is used to compute the gradient operators in this paper as shown in Figure 2. Details about this algorithm can be found in reference [27]. After Δ*x* and Δ*y* are obtained using Equation (4), the total displacement for point *P* along *x* and *y* direction are *u* + Δ*x* and *v* + Δ*y*, respectively.

## 3. Double Precision Gradient-Based Method for Subpixel Matching

### 3.1. Gradient-Based Subpixel Matching Method

Although the gradient-based method has the advantage of high efficiency for subpixel matching, Pan pointed out that the accuracy should be improved using additional steps [26], especially when the true subpixel displacement is far away from the interpolation center. This phenomenon is related to accuracy reduction using an extrapolation technique since the accurate *g_x_* and *g_y_* in Equation (4) can only be acquired around the interpolation center. To solve this problem, the best matching integral pixel point is selected as the center for the bicubic spline interpolation. However, the accuracy of the calculated subpixel displacement will decrease unacceptably when the subpixel distance is farther than 0.5 pixel. This phenomenon can be explained using the following numerical simulation.

An 8-bit speckle pattern with a resolution of 256 × 256 pixels was obtained using the iterative spatial-gradient based algorithm [20], as shown in Figure 3a. Ten translated images were generated with an equal distance in *x* direction, corresponding to a shift of 0.1 pixels between two successive speckle images. Thus, only subpixel displacement extraction in *x* direction was considered here. A total of 576 (24 × 24) points located in the intersection of grid of 10 pixels were chosen to calculate subpixel displacement, that is, the distance between each point was 10 pixels, as shown in Figure 3a. Moreover, two adjacent integral pixels (here 0 represents the left pixel and 1 denotes right pixel) to each point were employed as the interpolation center, and then subpixel displacements corresponding to all 576 points were extracted using the traditional gradient-based (GB) method. The mean error and standard variance of subpixel displacement calculated using 576 points at each deformed image were plotted in Figure 4. The dots represent the mean error, and the upper and lower bars represent the 90% deviation lines. It was shown that mean error would increase significantly when the distance between the actual subpixel displacement and the interpolation center exceeded 0.5 pixel. This trend is directly related to the first-order derivatives used in Equation (4), which are accurate only in the vicinity of the interpolation center. Meanwhile, the standard deviation error has the same trend, which will increase from 0.003 to 0.007 when the distance between the actual subpixel position and interpolation center is near or beyond 0.5 pixel.

Furthermore, images may be contaminated by a variety of noise sources, such as photon noise, thermal noise, readout noise, and shot noise [28]. A high noise level will lead to unacceptable errors using the gradient-based method. To practically simulate random noise occurring in real situations, Gaussian noises with zero mean and five different standard deviations (varying from 1 to 5) [26] were added into the first image set (noiseless image) to generate five other image sets. The image with standard deviation (SD) 5 is depicted in Figure 3b. For all five sets of images with various noise levels, the same operation procedure in the noise-free case was implemented and results are shown in Figure 5.

As can be seen from Figure 5, mean errors and standard deviations increased with the augment of noise variance levels. It can be concluded that the noise in the image must be considered to improve the subpixel matching accuracy.

### 3.2. Double Precision Gradient-Based (DPG) Method

To calculate the subpixel displacement, the GB method uses the nearest integral pixel as the center for the five-point bicubic spline interpolation, and reasonable accuracy will be acquired when the appropriate integer pixel is chosen. Unfortunately, it is impossible to take the appropriate integral pixel as the center every time. Therefore, these two near neighborhood integer pixels along one direction are used to calculate subpixel displacement associated with white noise, and then the identified displacement is linearly combined via a weight coefficient to obtain the final identification results. Only the details in the *x* direction are described in the following part since the subpixel matching is recognized in *x* and *y* directions under 2D measurement conditions.

Firstly, the subpixel displacements Δ*x*_0_ and Δ*x*_1_ were extracted with the traditional GB method using the left and right pixel as an interpolation center, respectively. Then, the final subpixel displacement Δ*x* was obtained by
(5)$$\Delta x=(1-a_{SD,d})\Delta x_{0}+a_{SD,d}\Delta x_{1}$$
where *a_SD,d_* represents the weight coefficient, subscript *d* denotes the subpixel position and *SD* corresponds to the noise variance.

From Figure 4a, the mean error is very small when the true subpixel displacement is closer to the interpolation center, that is, Δ*x* should be equal to Δ*x*_0_ if the true subpixel displacement is 0 and the left pixel is used as interpolation center. Correspondingly, Δ*x* should be equal to Δ*x_1_* when the true subpixel displacement is 1 and the right pixel is utilized as interpolation center. At the same time, the mean errors calculated from either interpolation center were almost equal when the true subpixel displacement is round 0.5 pixel, so the weight for Δ*x*_0_ and Δ*x*_1_ should be equal. Therefore, it is easy to draw the conclusion that *a_SD,_*_0_ = 0, *a_SD,_*_1_ = 1 and *a_SD,_*_0.5_ = 0.5.

Then, the discrete point *a_SD,d_* was obtained by substituting the data in Figure 5 into Equation (5), and results from the first two images, where the true subpixel displacement is 0.1 and 0.2 pixel respectively, are plotted in Figure 6. According to the distribution rule of discrete points *a_:,d_* (denotes the *a_SD,d_* under different variance of noise) in Figure 6, it is known that the weight coefficient *a_:,d_* first increases with the variance and then tends to be stable. Consequently, a formula in exponential form was adopted to fit the parameter *a_:,d_* for a fixed subpixel displacement:(6)$$a_{:,d}=c_{1}e^{-c_{2}\cdot SD}+c_{3}$$
where *c*_1_, *c*_2_, and *c*_3_ are fitting parameters and *SD* represents the variance of noise. It should be noted that we assumed that the noise level was the same for the reference image and a series of deformed images. For example, these factors, such as lighting and noise, remained unchanged during the measurement procedure. Furthermore, parameters in Equation (6) for other *a_:,d_* were obtained and are listed in Table 1 using the same operation. 

Therefore, the parameter *a_SD,d_* in Equation (5) was further obtained by fitting using the spline curve as,
(7)$$a_{SD,d}=spline\left[a_{:,d}\right]$$

Figure 7 shows the curved surface of the weight coefficient *a_SD,d_* under different *SD* and the pre-assigned displacement at decile points between 0 and 1. Extracting parameter *a_SD,d_* from this surface, the subpixel displacement Δ*x* could then be calculated using Equation (5).

The detailed procedure of the proposed DPG method is as follows: 

**Step 1:** Calculate the noise variance *SD*. Before the test, two consecutive images under the test condition are taken and the noise variance *SD* is obtained by calculating the mean square error of the gray-scale intensity of these two consecutive images. 

**Step 2:** Determine Δ*x*_0_ and Δ*x*_1_. For the subpixel matching, as a point *P*’ in the deformed image is selected, the subpixel displacement Δ*x_t_* is computed using the traditional GB method. If Δ*x_t_* is less than 0.5 pixels, replace Δ*x_t_* with Δ*x*_0_. Then, taking the integral pixel at the right of the point *P*’ as the interpolation center, the subpixel displacement Δ*x*_1_ is further extracted using the GB method; if Δ*x_t_* is greater than 0.5 pixels, replace Δ*x_t_* with Δ*x*_1_. Taking the integral pixel at the left of the point *P*′ as the interpolation center, the subpixel displacement Δ*x*_0_ is then computed using the GB method.

**Step 3:** Determine the *d*. Calculate *d* using *d* = min(|Δ*x*_0_|, |1 − Δ*x*_1_|).

**Step 4:** Calculate the weight coefficient *a_SD,d_*. The *a_:,d_* was obtained by Equation (6) according to the *SD* and parameters *c_j_* (*j* =1,2,3) in Table 1. Substitute the *a_:,d_* into Equation (7) and determine the weight coefficient *a_SD,d_* using the subscript *SD* and *d*.

**Step 5:** Calculate the subpixel displacement Δ*x*. Substitute *a_SD,d_*, Δ*x*_0_, and Δ*x*_1_ into Equation (5) to get the identification subpixel displacement Δ*x*.

Subpixel displacement in the *y* direction can be calculated using the same procedure, so it will not be described in detail here. 

## 4. Experimental Verification

### 4.1. Numerical Simulation Using Speckle Pattern

To verify the proposed DPG method, a speckle pattern with size 256 × 256 pixels and 256 gray levels was simulated with the iterative spatial gradient-based algorithm, as shown in Figure 8a. The speckle granule was set as 2000, and the radius of each speckle granule was 3 pixels with 0.7 peak intensity. A unified coordinate system *x*-*y* was established for the speckle pattern and the section inside the red wireframe with 250 × 250 pixels was set as the region of interest. The simulated speckle pattern was employed as the reference image. Deformed images were copied from the reference one with random shift from 0.1 to 1 pixel in the *x* direction since the purpose was only to verify the subpixel matching. It should be noted that only in-plane translation was performed for deformed images in the numerical example, and the maximum translation was 1 pixel, so that the integral pixel search algorithm dose was not used in the numerical simulation. Moreover, to simulate the photography noise from camera, additive random Gaussian noise with zero mean and random variance from 1 to 5 was added to each shifted image. Here, randomly selected speckle images with different noise variances are expressed using PSNR, as shown in Figure 8b.

The proposed DPG, GB, and N–R method were performed on a desk computer (4 cores of Inter(R) Core (TM) i5 CPU750 with 2.67GHz main frequency) and the computational tool was MATLAB on the computer. The subset size was set as 41 × 41 pixels, and the mean errors with random standard deviation for those three algorithms are given in Figure 9.

As seen in Figure 9, the maximum error of the GB method was close to 0.01 pixel, while the error of the improved DPG was less than 0.0025 pixel, which is equivalent to that of the N–R method. This is because the DPG method not only adopts weight coefficient to consider the two nearest integral pixels to improve accuracy, but also covers the influence of noise on the recognition results. 

Computation times for the three different algorithms are listed in Table 2. It should be noted that computation efficiency depends on a number of factors, such as programming efficiency, usage of program language, the performance of hardware, etc. The relative performance, however, can be demonstrated. 

Table 2 shows that the GB method was the most efficient way where a reliability-guided displacement scanning strategy was employed to avoid the time-consuming displacement searching. Moreover, a pre-computed global interpolation coefficient look-up table using bicubic spline interpolations was employed to eliminate repetitive interpolation calculation. Furthermore, the computational efficiency of the DPG method was approximately twice longer than that of the GB method. This is because the proposed method uses one more interpolation calculation than the GB method. However, the computation cost of the N–R method is 13.7 times that of the proposed DPG method, that is to say, the calculation efficiency of the proposed method was 92.67% higher than that of the Newton–Raphson method, indicating that the N–R method was prohibited when dynamic displacement needed to be identified from an ocean of images.

### 4.2. Painted Beams Experiment

#### 4.2.1. Experiment Setup

To further verify the proposed method, a simply supported beam with Chinese ancient painting on the flank was implemented, as shown in Figure 10 and Figure 11. The length, height, and width of the beam were 1400, 100, and 50 mm, respectively. The concentrated load was applied using a two-point distribution beam under a third-point bending of the tested beam. Three micrometers were used to measure deflection on quarter positions of the beam and the acquired data were taken as the reference displacement. 

A DIC measurement system with charge-couple devices (CCD) camera (model GZL-CL-41C6M-C with 2048 × 2048 pixels resolution) was placed on a tripod about 4 m in front of the painted beam. Its optical axis was aligned to the shape center of the beam and perpendicular to the painted pattern surface. It was assumed that the optical axis of the camera was perpendicular to the plane of the measured object. Images before and after the loading procedure with a size of 2048 × 2048 pixels at 8-bit gray levels recorded through the CCD camera. In order to confirm the loading equipment and instrument, preloading was performed before formally loading, and the noise variance could be obtained using the CCD camera after setting up the experimental environment. 

#### 4.2.2. Experiment Results

To compare with test values obtained by these micrometers, only the area marked with a red dot line near the micrometers is considered in this subsection, as shown in Figure 11b. Coordinates of these selected pixel points from top to bottom of the beam were set as 1–168 pixel, and the corresponding distance measured with the micrometer was 100 mm. Therefore, the calibration factor was 100/168 = 0.5952 mm/pixel for the experiment system. It should be noted that the coarse-fine search method was adopted in the integer pixel search step both for the GB and DPG methods. When using the algorithmic process, directly using the captured images for processing may be affected by uncertainty and/or inaccuracy. At this time, you can choose to use techniques based on fuzzy logic methods [29,30]. These methods are very suitable for the preprocessing of images affected by performance and/or imprecision. Although this technology was not used in this study, these technologies can be considered in any future development. Taking the mid-span displacement of the beams as a reference, 5 loading cases were implemented, as shown in Table 3.

Taking the displacement from the micrometer as true displacement, the GB and DPG method were used to identify the displacement under different test conditions. The identification results are shown in Table 4, Table 5 and Table 6. The mean error of the deflection identification results of the DPG method was generally less than 0.1 mm, and the maximum absolute error was 0.083 mm, while the mean error of the deflection identification results of the GB method was generally above 0.1mm, and the maximum absolute error was 0.554 mm. It can be concluded that the accuracy of the DPG method was superior to that of the GB method. Moreover, the standard deviation obtained from the GB method was generally larger than the DPG method, which shows that the DPG method has better noise stability.

Identification results along the height of the wooden beam under case 5 are depicted in Figure 12. It can be clearly seen that displacement identified by GB method differed greatly from those identified by the other two methods, and the result fluctuates up and down all the time, indicating that the accuracy of the GB method is not high and the stability is poor. The DPG method not only achieved high accuracy as that of the N–R method, but it also has good stability. At the mid-span monitoring position near the top of the beam, the identified deflection may produce a small error. This is because the image recognition will be affected by the background image when the boundary of pixel subsets contacts the edge of the wood beam or exceeds the edge of the wood beam, so recognition results may fluctuate. Therefore, when the displacement is monitored by the DIC method, some special algorithms should be utilized to address the boundary registration, which is beyond the scope of this study.

#### 4.2.3. Light Condition Analysis

Some studies have reported that light variation has a great influence on the accuracy and stability of identification results for DIC method [31]. In order to exploit the influence of light condition on the accuracy of the proposed method, the light conditions were set as light and dark groups, and their grayscale image is shown in Figure 13. The experimental setup in Section 4.2.1 was also used here. Because the experiment was carried out in a laboratory environment with lights, the experiment at that time was defined as the light group and the experiment done with the light in the laboratory turned off was defined as the dark group.

Since the trend was similar for all cases, only the results at micrometer 2 under case 5 are depicted in Figure 14, and Figure 15 shows the mean error of the recognition results under two groups of light and dark using different algorithm. Combining Figure 14 and Figure 15, it can be seen that in the “light” group with better lighting conditions, the mean error of the recognition results using GB, DPG, and N–R methods was almost the same. Meanwhile, in the “dark” group with poor lighting conditions, the deflection acquired by the DPG and N–R method fluctuated slightly along the identification area, but still maintained relatively high precision, while identification results by the GB method fluctuated significantly, indicating that the accuracy of the GB method is greatly dependent on the light condition.

## 5. Conclusions

The DPG algorithm is proposed in this study for fast and accurate displacement extraction for DIC subpixel matching. The grayscale of the two nearest integral pixels was taken as the center point for the five-point bicubic spline interpolation. Then, subpixel displacement was obtained from these two integral pixels using the traditional gradient-based method. In order to improve accuracy and anti-noise capability, a method using linear combination strategy was presented and the procedure to calculate a weight coefficient was introduced in detail. Consequently, the final subpixel displacement was achieved by linear merging these identification results from the traditional GB method. A simulation speckle pattern and a painted beam experiment were implemented to verify the practicability of the proposed method. Results show that the accuracy of the proposed method for recognizing subpixel displacement was improved by 4 times compared with the GB method. Meanwhile, the computational efficiency of the proposed DPG method was at least 13.7 times higher than that of the traditional N–R method, which provides the possibility for real-time dynamic displacement measurement for an ocean of images. Furthermore, the proposed method had better anti-noise performance and stability than the GB method under different light conditions. This merit will greatly reduce the impact of the environment on identification accuracy in the actual experiment. This DPG algorithm can be considered a complement to target tracking and displacement measurement.

The proposed DPG method also has certain limitations, such as calculation speed and stability, under certain noise conditions. Further study will be focused on exploiting other types of noise, such as salt and pepper noise, and large deformation cases where the non-translation subset should be considered.

## Figures and Tables

**Figure 1 sensors-21-03140-f001:**
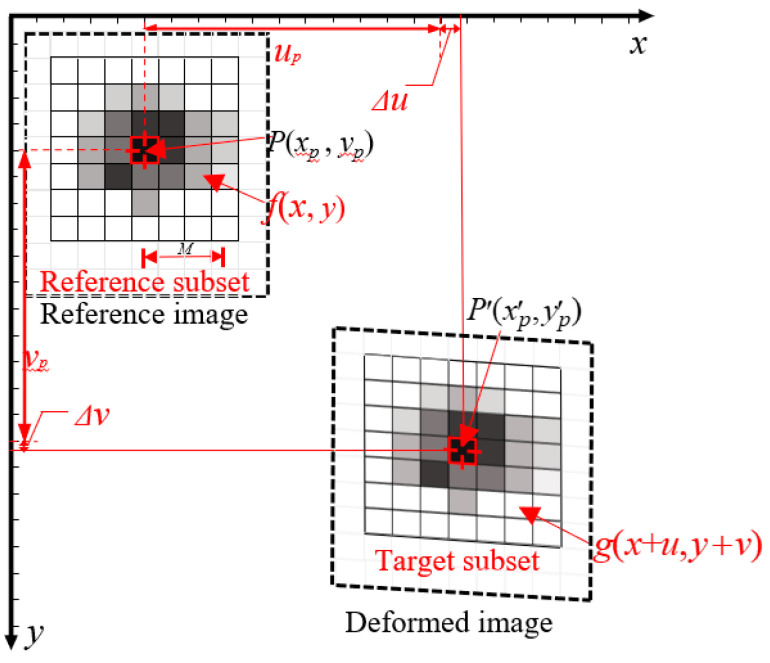
Displacement identification principle of two-dimensional DIC during integral pixel step.

**Figure 2 sensors-21-03140-f002:**
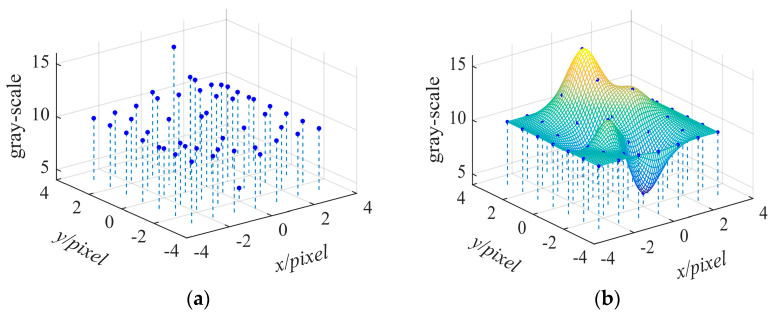
Bicubic spline interpolation diagram: (**a**) the discrete grayscale points of integral pixel; (**b**) the bicubic spline interpolated graph. Notes: The negative pixel in this figure does not mean that it is absolutely negative, but only represents relative position.

**Figure 3 sensors-21-03140-f003:**
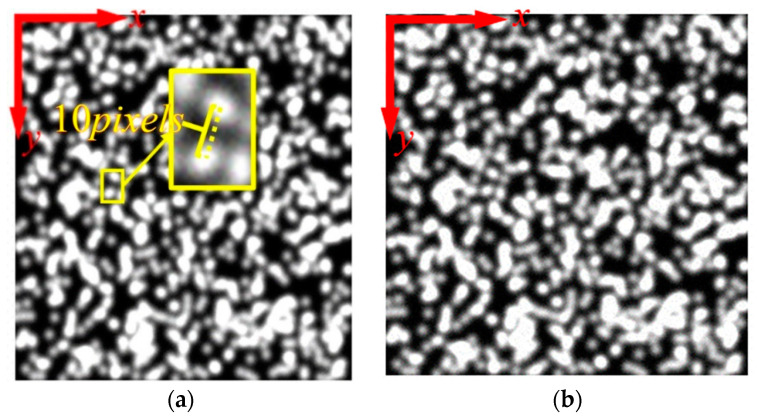
Speckle pattern: (**a**) without noise; (**b**) Gaussian white noise with zero mean and variance 5.

**Figure 4 sensors-21-03140-f004:**
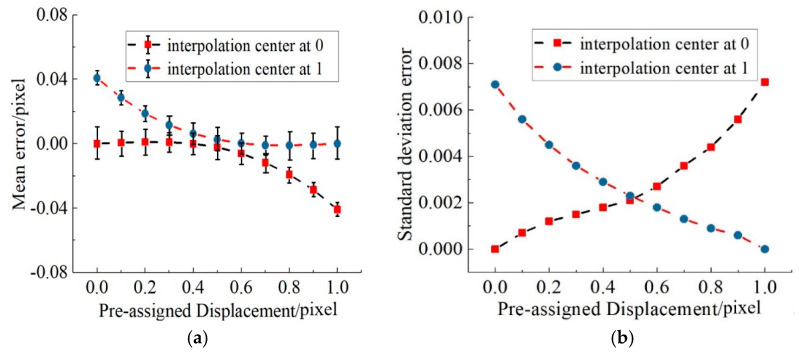
Matching error without noise: (**a**) mean error; (**b**) standard deviation error.

**Figure 5 sensors-21-03140-f005:**
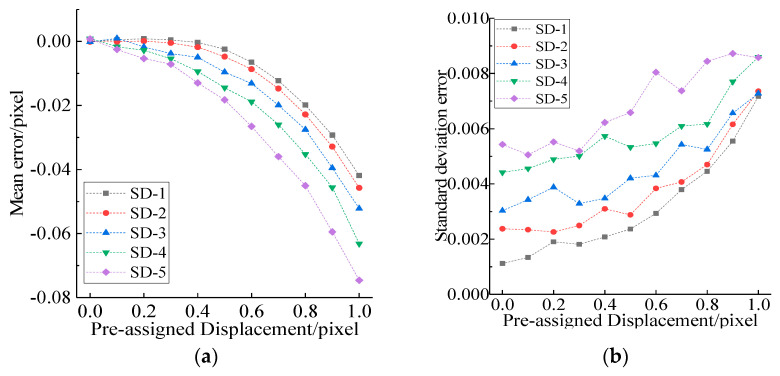
Matching error with noise and the interpolation centered at 0 pixel: (**a**) mean error; (**b**) standard deviation error.

**Figure 6 sensors-21-03140-f006:**
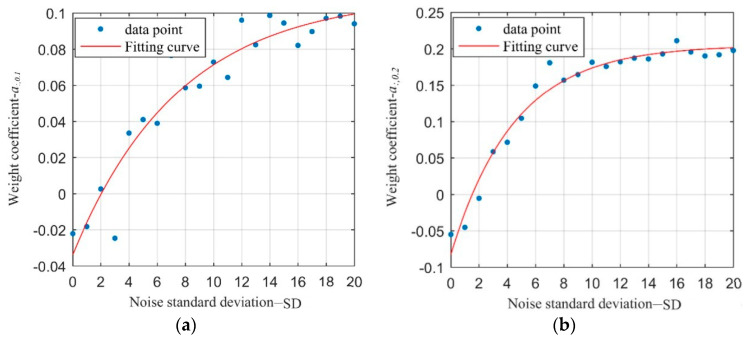
Fitting diagram of noise *SD* and weight coefficient *a_:,d_*: (**a**) *a_:,_*_0.1_; (**b**) *a_:,_*_0.2_.

**Figure 7 sensors-21-03140-f007:**
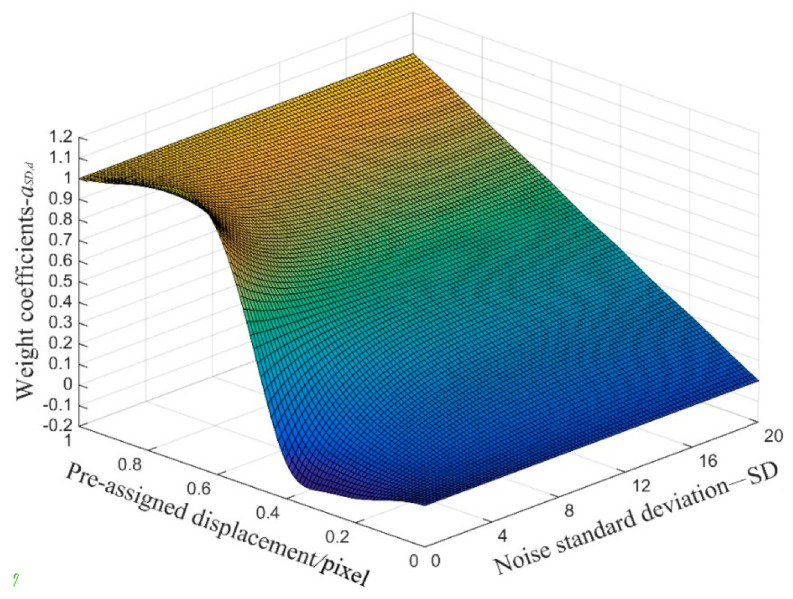
The curved surface of weight coefficient *a_SD,d_*.

**Figure 8 sensors-21-03140-f008:**
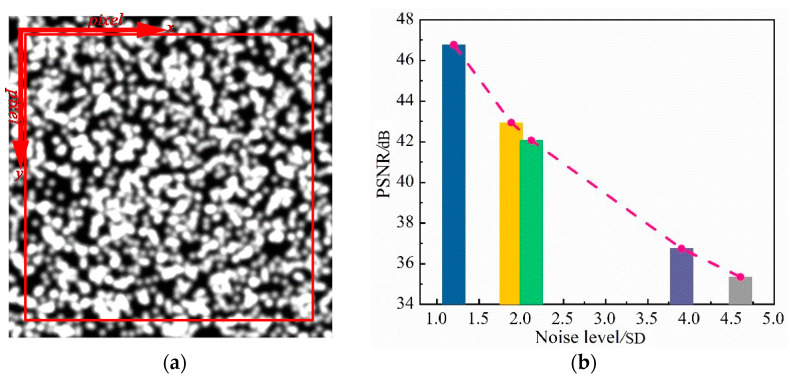
(**a**) Speckle pattern of the reference image used in numerical simulation; (**b**) PSNR under randomly selected noise.

**Figure 9 sensors-21-03140-f009:**
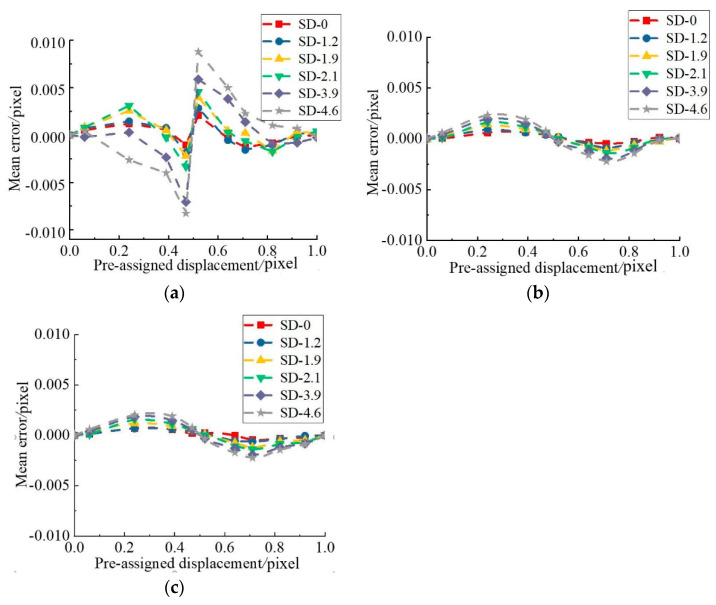
Mean errors of displacements measured with different noise level by (**a**) GB, (**b**) DPG, and (**c**) N–R method.

**Figure 10 sensors-21-03140-f010:**
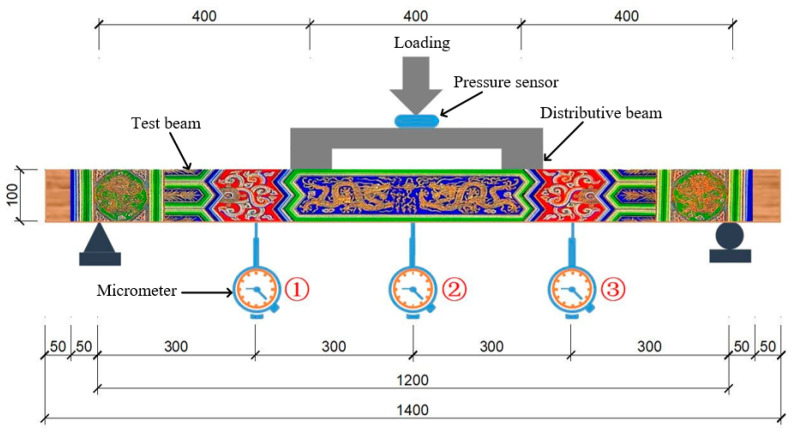
The schematic diagram of the experiment (units: mm).

**Figure 11 sensors-21-03140-f011:**
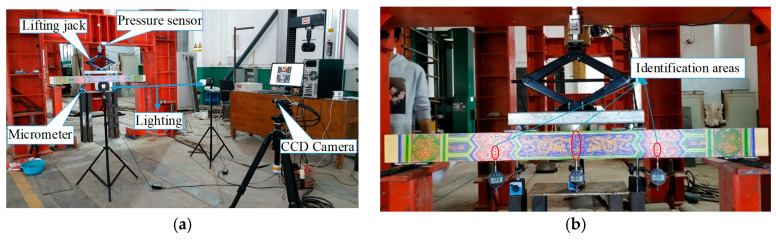
The testing site with panorama: (**a**) experiment setup; (**b**) the test beam.

**Figure 12 sensors-21-03140-f012:**
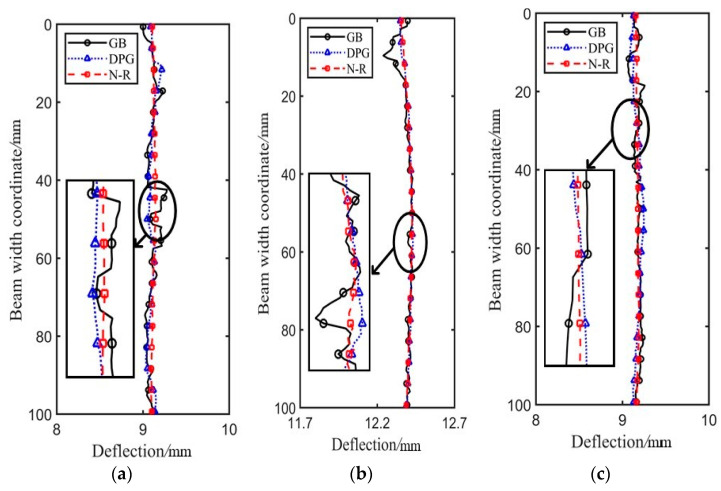
Identification results along the height of beam: (**a**) Micrometer 1; (**b**) Micrometer 2; (**c**) Micrometer 3.

**Figure 13 sensors-21-03140-f013:**
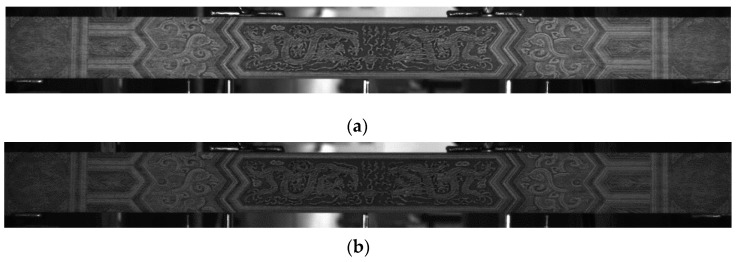
Grayscale images of the painting beam: (**a**) light group; (**b**) dark group.

**Figure 14 sensors-21-03140-f014:**
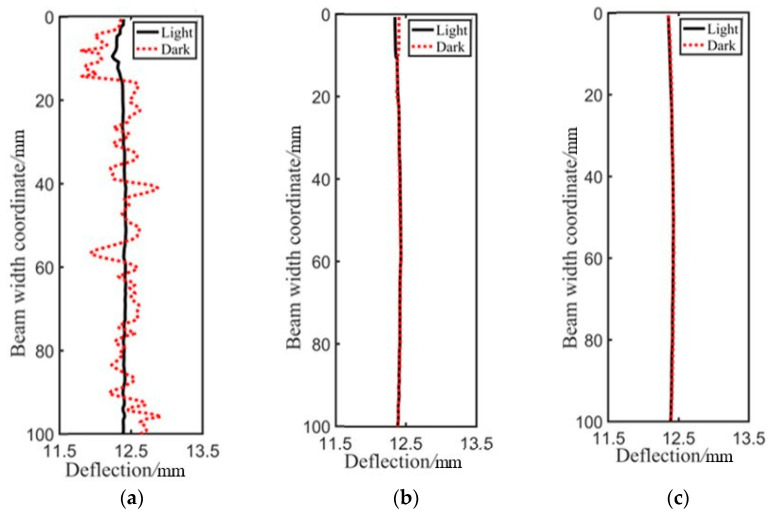
Identification results with different lighting conditions at micrometer 2: (**a**) GB; (**b**) DPG; (**c**) N–R.

**Figure 15 sensors-21-03140-f015:**
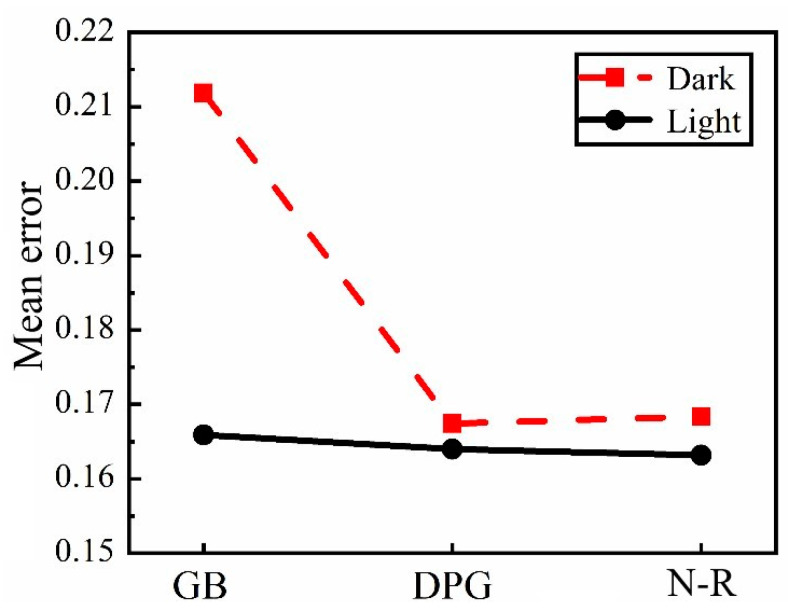
Mean error of recognition results under different algorithms.

**Table 1 sensors-21-03140-t001:** Coefficient for *a_:,d_*.

*d*	0.0	0.1	0.2	0.3	0.4	0.5	0.6	0.7	0.8	0.9	1.0
*c_1_*	0.000	−0.143	−0.287	−0.403	−0.432	0.000	0.375	0.411	0.282	0.133	0.000
*c_2_*	0.000	−0.132	−0.222	−0.271	−0.362	0.500	−0.370	−0.307	−0.240	−0.170	0.000
*c_3_*	0.000	0.110	0.205	0.298	0.403	0.500	0.608	0.704	0.803	0.901	1.000

**Table 2 sensors-21-03140-t002:** Computation efficiency comparison.

Method	GB	DPG	N–R
Compution time (s)	0.404	0.827	11.287

**Table 3 sensors-21-03140-t003:** Beam deflection under different cases (units: mm).

Loading Case	1	2	3	4	5
Deflection at mid-span	2.4	4.8	7.2	9.6	12

**Table 4 sensors-21-03140-t004:** Deflection identification at the micrometer 1 (units: mm).

Case	Micro-Meter	GB			DPG		
		Identification	Mean Error	Standard Deviation	Identification	Mean Error	Standard Deviation
1	1.979	1.916	−0.063	1.050	1.971	−0.008	0.010
2	3.735	3.691	−0.044	1.984	3.724	−0.011	0.015
3	5.467	5.302	−0.165	2.508	5.457	−0.010	0.024
4	7.181	6.962	−0.219	0.894	7.178	−0.003	0.028
5	8.936	8.719	−0.217	0.384	8.930	−0.006	0.040

**Table 5 sensors-21-03140-t005:** Deflection identification at the micrometer 2 (units: mm).

Case	Micro-Meter	GB			DPG		
		Identification	Mean Error	Standard Deviation	Identification	Mean Error	Standard Deviation
1	2.414	2.573	0.159	0.016	2.475	0.061	0.012
2	4.828	4.717	−0.111	2.038	4.911	0.083	0.015
3	7.214	6.797	−0.417	2.915	7.289	0.075	0.018
4	9.602	9.048	−0.554	3.099	9.626	0.024	0.037
5	12.012	11.465	−0.547	0.116	11.969	−0.043	0.042

**Table 6 sensors-21-03140-t006:** Deflection identification at the micrometer 3 (units: mm).

Case	Micro-Meter	GB			DPG		
		Identification	Mean Error	Standard Deviation	Identification	Mean Error	Standard Deviation
1	1.972	2.047	0.075	0.441	1.922	−0.050	0.021
2	3.702	3.774	0.072	2.418	3.634	−0.068	0.030
3	5.420	5.617	0.197	0.982	5.359	−0.062	0.041
4	7.132	6.871	−0.261	0.447	7.112	−0.020	0.050
5	8.860	8.602	−0.258	0.298	8.825	−0.035	0.059

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
