# Peer review of "Subpixel Matching Using Double-Precision Gradient-Based Method for Digital Image Correlation"

_sensors, 2021, doi:10.3390/s21093140_

Round 1

Reviewer 1 Report

The double-precision gradient-based algorithm is proposed in this study for fast and accurate displacement extraction for digital image correlation subpixel matching. The grayscale of these two nearest integral pixels is taken as the center point for the five-point bicubic spline interpolation. And then subpixel displacements are obtained from these two integral pixels using the traditional gradient-based method. In order to improve accuracy and anti-noise capability, a method using linear combination strategy is presented and the procedure to calculate weight coefficient is introduced in detail. Consequently, the final subpixel displacement is achieved by linear merging these identification results from the traditional gradient-based method. The paper is interesting but has drawbacks:

  1. The motivation of the paper is not clearly stated. 
  2. The point with the levels of noise is not clear. The authors should present noise in terms of PSNR.
  3. Table 2 with performance comparison should be better commented on.
  4. Subpart 4.2.3 is not clear and should be expanded. The relationship between a light condition and the quality of the proposed solution should be presented.
  5.  The paper should be proofread.

Author Response

Responds to the reviewer’s comments:

Reviewer 1:

1. Response to comment: (The motivation of the paper is not clearly stated)

Response: It is really true as Reviewer suggested that the motivation of this paper “This paper is motivated by this feature of the GB method to try to find a sub-pixel algorithm that can inherit the high efficiency of the GB method while having the characteristics of high precision” has been supplemented on line 69 and marked in red.

2. Response to comment: (The point with the levels of noise is not clear. The authors should present noise in terms of PSNR)

Response: PSNR is the ratio of the maximum possible power of a signal to the destructive noise power that affects its representation accuracy. It is often simply defined by the mean square error (MSE).

PSNR evaluates the degree of agreement between two images. However, according to the error analysis theory, the calculation error of the sub-pixel displacement measurement algorithm is composed of the mean error and the standard deviation. Therefore, the average error and standard deviation used in the article to evaluate the absolute error of a sub-pixel displacement measurement algorithm. At the same time, this paper refers to a number of different articles evaluating sub-pixel noise, as follows,

1). Pan B , Li K , Tong W . Fast, Robust and Accurate Digital Image Correlation Calculation Without Redundant Computations. Experimental Mechanics, 2013, 53(7):1277-1289. doi:10.1007/s11340-013-9717-6

2). Pan, B. Full-field strain measurement using a two-dimensional Savitzky-Golay digital differentiator in digital image correlation. Opt. Eng, 2007, 46(3):033601. doi: 10.1117/1.2714926

3). Wang Z Y , Li H Q , Tong J W , et al. Statistical Analysis of the Effect of Intensity Pattern Noise on the Displacement Measurement Precision of Digital Image Correlation Using Self-correlated Images. Experimental Mechanics, 2007, 47(5):701-707. doi: 10.1007/s11340-006-9005-9

4). Pan B , Xie H , Wang Z , et al. Study on subset size selection in digital image correlation for speckle patterns. Optics Express, 2008, 16(10):7037-7048. doi: 10.1364/OE.16.007037

5). Schreier, Hubert W . Systematic errors in digital image correlation caused by intensity interpolation. Optical Engineering, 2000, 39(11):2915-2921. doi:10.1117/1.1314593

6).Pan B, Xie H, Dai F. An investigation of sub-pixel displacements registration algorithms in digital image correlation. Chin J Theor Appl Mech, 2007, (02):103-110. doi:10.3321/j.issn:0459-1879.2007.02.014

3. Response to comment: (Table 2 with performance comparison should be better commented on)

Response: It is really true as Reviewer suggested that the comment “the calculation efficiency of the proposed method is 92.67% higher than that the Newton-Raphson method” in Table 2 have been added on line 69 in the paper and marked in red.

4. Response to comment: (Subpart 4.2.3 is not clear and should be expanded. The relationship between a light condition and the quality of the proposed solution should be presented)

Response: We have supplemented the relationship between a light condition and the quality of the proposed solution “The experiment setup in section 4.2.1 is still used here. Because the experiment was carried out in the laboratory environment with lights, the experiment at this time was defined as the light group and the experiment done with the light in the laboratory turned off is defined as the dark group” according to the Reviewer’s suggestion on line 405 in the paper and marked in red.

5. Response to comment: (The paper should be proofread)

Response: We have carried out a detailed and rigorous proofread of the entire article.

Other changes:

1. Line 10, the statements of “a series images in the DIC approach” were corrected as “a series of images in the DIC approach”

2. Line 14, the statements of “the traditional gradient method is used to…” were corrected as “the traditional gradient-based method is used to…”

3. Line 16, the statements of “…direction are both utilized as interpolation center” were corrected as “…direction are both utilized as an interpolation center”

4. Line 44, the statements of “and performance of these methods been…” were corrected as “and the performance of these methods has been…”

5. Line 66, the statements of “The accuracy of GB method…” were corrected as “The accuracy of the GB method…”

6. Line 74, the statements of “whose accuracy are determined by center selection” were corrected as “whose accuracy is determined by center selection”.

7. Line 81, the statements of “then the identified displacement is linear combined” were corrected as “then the identified displacement is linearly combined”.

8. Line 83, the statements of “but also improves accuracy…” were corrected as “but also improves the accuracy…”.

9. Line 189, the statements of “lower bars represents the 90% deviation lines” were corrected as “lower bars represent the 90% deviation lines”.

10. Line 338, the statements of “height and wide of the beam is 1400,100 and 50 mm respectively” were corrected as “height and width of the beam are 1400,100 and 50 mm respectively”.

11. Line 347, the statements of “at 8-bit gray levels were record through…” were corrected as “at 8-bit gray levels recorded through…”.

12. Line 350, the statements of “camera after setting up the experimental environmental” were corrected as “camera after setting up the experimental environment”.

13. Line 358, the statements of “with red dot line near the micrometers is considered” were corrected as “with a red dot line near the micrometers is considered”.

14. Line 386, the statements of “displacements identified by GB method differs…” were corrected as “displacements identified by GB method differ…”

15. Line 389, the statements of “The DPG method not only achieve high accuracy…” were corrected as “The DPG method not only achieves high accuracy…”

16. Line 396, the statements of “beyond the scope this study” were corrected as “beyond the scope of this study”

17. Line 419, the statements of “indicating accuracy of the GB method is greatly depend on” were corrected as “indicating accuracy of the GB method is greatly dependent on”

18. Line 468-471, “Baqersad J, Poozesh P, Niezrecki C, et al. Photogrammetry and optical methods in structural dynamics–A review. Mechanical Systems and Signal Processing, 2016, S0888327016000388–1:18. doi:10.1016/j.ymssp.2016.02.011 

Feng D, Feng M Q. Computer vision for SHM of civil infrastructure: From dynamic response measurement to damage detection–A review. Engineering Structures, 2018, 156(FEB.1):105-117. doi:10.1016/j.engstruct.2017.11.018” was added  

19. Line 477-485, “Chen M, Tang Y C, Zou X, et al. High-accuracy multi-camera reconstruction enhanced by adaptive point cloud correction algorithm. Optics and Lasers in Engineering, 2019, 122:170-183. doi:10.1016/j.optlaseng.2019.06.011

Tang Y, Li L, Wang C, et al. Real-time detection of surface deformation and strain in recycled aggregate concrete-filled steel tubular columns via four-ocular vision. Robotics and Computer-Integrated Manufacturing, 2019, 59: 36-46. doi: 10.1016/j.rcim.2019.03.001

Tang Y, Chen M, Lin Y, et al. Vision-Based Three-Dimensional Reconstruction and Monitoring of Large-Scale Steel Tubular Structures. Advances in Civil Engineering, 2020, 1–17. (Prepublish) doi:10.1155/2020/1236021

Rizo-Patron S , Sirohi J . Operational Modal Analysis of a Helicopter Rotor Blade Using Digital Image Correlation. Experimental Mechanics, 2016, 57(3):1-9. doi:10.1007/s11340-016-0230-6” was added.  

20. Line 508-509, “BING P, HUIMIN X, ZHAO Y. Equivalence of digital image correlation criteria for pattern matching. Applied Optics, 2010,49(28):5501. doi:10.1364/AO.49.005501” was added.

21. Line 523-526, “Orujov F, Maskeliunas R, Damasevicius R, et al. Fuzzy based image edge detection algorithm for blood vessel detection in retinal images. Applied Soft Computing, 2020, 94:106452. doi: 10.1016/j.asoc.2020.106452

Versaci M, Morabito F C. Image Edge Detection: A New Approach Based on Fuzzy Entropy and Fuzzy Divergence. International Journal of Fuzzy Systems, 2021,1-19. (Prepublish) doi: 10.1007/s40815-020-01030-5” was added.

We tried our best to improve the manuscript and made some changes in the manuscript. These changes will not influence the content and framework of the paper. And here we did not list the changes but marked in red in revised paper.

We appreciate for Editors/Reviewers’ warm work earnestly and hope that the correction will meet with approval.

Once again, thank you very much for your comments and suggestions.

Reviewer 2 Report

This manuscript introduces a double-precision gradient-based image matching algorithm for sub-pixel matching in a noisy environment. The research is interesting and provides valuable results, but the current document has several weaknesses that must be strengthened in order to obtain a documentary result that is equal to the value of the publication.

General considerations:

(1)At the thematic level, the proposal provides a very interesting vision, as a high-precision image matching method would be a very useful resource for engineers.

(2)Concerning the presentation of the contents, the document is acceptable. Nonetheless, it is recommended that authors develop proofreading to avoid common mistakes such as word confusion, incorrect expressions, continuous repetition of the same words and expressions, incorrect use of punctuation rules, words without a space between them, etc.

(3)The document contains a total of 22 employed references, of which 8 are publications produced in the last 5 years (36%), 3 in the last 5-10 years (14%), 11more than 10 years old (50%), implying a total percentage of 50 % recent references. In this way, the number of recent references is in More state-of-the-art researches should be introduced in this section.

(4)We all know that the optical flow method is a classic and effective method to complete target tracking. Please briefly explain how the method proposed in this article compares with the classical optical flow method? Where is the advantage of the proposed method?

(5)Strictly speaking, only when the optical axis of the camera is perpendicular to the plane of the measured object, the proportion of the pixel length and the physical length is fixed.  Is your manuscript based on such assumptions when obtaining experimental data?

(6) Did you only use a set of experiments to evaluate the effect of the algorithm under different lighting? How to objectively define the "dark" and "bright" of the experimental object?

(7) Nonlinear light is the most common factor that affects the performance of image processing algorithms. It is recommended to supplement the experiment under this type of lighting condition.

Title, Abstract, and Keywords:

(8) The abstract is complete and well-structured and explains the contents of the document very well. Nonetheless, the part relating to the results could provide numerical indicators obtained in the research.

Chapter 1: Introduction

(9) The paragraphs introducing the research topic give a too simple view of the problems related to your topic and should be revised and completed with citations to authority references (High-accuracy multi-camera reconstruction enhanced by adaptive point cloud correction algorithm;  Real-time detection of surface deformation and strain in recycled aggregate concrete-filled steel tubular columns via four-ocular vision).

(10) The novelty of the study is not apparent enough. In the introduction section, please highlight the contribution of your work by placing it in context with the work that has done previously in the same domain.

(11) Vision technology, including mark-free steoro vision method and DIC, is emerging these years. It should also be introduced for a full glance of the scope of related area. For field deformation detection, please refer to Vision-based three-dimensional reconstruction and monitoring of large-scale steel tubular structures.

Chapter 2 and 3: The method

(12) An overall technical flow chart is necessary.

(13) It appears to be no indication of the computational tools and software resources used to carry out the methods presented. These issues could be presented in a more orderly and clear manner.

Chapter 4: Experiments and results

(14) Figure 8does not seem to be very clear with regard to the characteristics of the

Chapter 5: Conclusions

(15) After all that has been read, this technique can be considered as a complement to target tracking and displacement measurement, but it certainly does not seem to be a substitute for this work in its current state.

(16) I recommend including the limitations regarding the consideration of the calculation speed and stability of the proposed matching algorithm.

Author Response

Responds to the reviewer’s comments:

Reviewer 2:

1. Response to comment: (At the thematic level, the proposal provides a very interesting vision, as a high-precision image matching method would be a very useful resource for engineers)

Response: Special thanks to you for your good comments.

2. Response to comment: (Concerning the presentation of the contents, the document is acceptable. Nonetheless, it is recommended that authors develop proofreading to avoid common mistakes such as word confusion, incorrect expressions, continuous repetition of the same words and expressions, incorrect use of punctuation rules, words without a space between them, etc.)

Response: We have carried out a detailed and rigorous proofread of the entire article.

3. Response to comment: (The document contains a total of 22 employed references, of which 8 are publications produced in the last 5 years (36%), 3 in the last 5-10 years (14%), 11more than 10 years old (50%), implying a total percentage of 50 % recent references. In this way, the number of recent references is in More state-of-the-art researches should be introduced in this section)

Response: We have supplemented the recent references with More state-of-the-art researches according to the Reviewer’s suggestion.

4. Response to comment: (We all know that the optical flow method is a classic and effective method to complete target tracking. Please briefly explain how the method proposed in this article compares with the classical optical flow method? Where is the advantage of the proposed method?)

Response: As Reviewer said that, the optical flow method is a classic and effective method to complete target tracking, and the optical flow method is divided into many types, such as image grayscale gradient-based, matching-based, phase-based, etc. The Double-Precision Gradient-based method in this paper is also a method based on image gradient-based, which is also a kind of optical flow method. The Double-Precision Gradient-based method proposed in this paper is an improvement on the traditional gradient-based method. After the improvement, compared with the original method, the calculation accuracy is greatly improved. Since the effect of noise is taken into account, the proposed method has higher anti-noise performance.

5. Response to comment: (Strictly speaking, only when the optical axis of the camera is perpendicular to the plane of the measured object, the proportion of the pixel length and the physical length is fixed.  Is your manuscript based on such assumptions when obtaining experimental data?)

Response: We are very sorry for neglecting to explain this assumption, the experiments in the paper are indeed based on this assumption and this assumption “It is assumed that the optical axis of the camera is perpendicular to the plane of the measured object” has been added on line 345 in the paper and marked in red.

6. Response to comment: (Did you only use a set of experiments to evaluate the effect of the algorithm under different lighting? How to objectively define the "dark" and "bright" of the experimental object?)

Response: As Reviewer said that this paper does only use a set of experiments to evaluate the effect of the algorithm under different lighting. This paper defines the “dark” and “bright” of the experimental object through the lighting conditions of the laboratory and this “The experiment setup in section 4.2.1 is still used here. Because the experiment was carried out in the laboratory environment with lights, the experiment at this time was defined as the light group and the experiment done with the light in the laboratory turned off is defined as the dark group” has been supplemented on line 405 in the paper and marked in red.

7. Response to comment: (Nonlinear light is the most common factor that affects the performance of image processing algorithms. It is recommended to supplement the experiment under this type of lighting condition)

Response: Considering that the pained beams in ancient buildings are irradiated by natural light, the light used in the experiment is natural light in the laboratory environment. For nonlinear light, it is not the scope of the thesis and subsequent research may consider the influences of nonlinear light.

8. Response to comment: (The abstract is complete and well-structured and explains the contents of the document very well. Nonetheless, the part relating to the results could provide numerical indicators obtained in the research.)

Response: As Reviewer suggested that we have supplemented “reaching the same high accuracy as the Newton-Raphson method, the calculation efficiently is 92.67% higher than that the Newton-Raphon method” the abstract on line 21 in the paper and marked in red.

9. Response to comment: (The paragraphs introducing the research topic give a too simple view of the problems related to your topic and should be revised and completed with citations to authority references (High-accuracy multi-camera reconstruction enhanced by adaptive point cloud correction algorithm; Real-time detection of surface deformation and strain in recycled aggregate concrete-filled steel tubular columns via four-ocular vision))

Response: We have re-written and supplemented the introduction according to the Reviewer’s suggestion on line 45 in the paper and marked in red.

10. Response to comment: (The novelty of the study is not apparent enough. In the introduction section, please highlight the contribution of your work by placing it in context with the work that has done previously in the same domain.)

Response: As Reviewer suggested that we have supplemented the work that has done previously in the same domain to highlight the contribution of this paper in the introduction.

11. Response to comment: (Vision technology, including mark-free steoro vision method and DIC, is emerging these years. It should also be introduced for a full glance of the scope of related area. For field deformation detection, please refer to Vision-based three-dimensional reconstruction and monitoring of large-scale steel tubular structures)

Response: As Reviewer suggested that we have supplemented the research in related fields such as vision technology in the introduction.

12. Response to comment: (An overall technical flow chart is necessary)

Response: As Reviewer said that section 3.2 of this paper has given the detailed step of DPG algorithm for the convenience of readers. Because all the steps have been given here, in order to reduce the burden, no more chart are draw.

13. Response to comment: (It appears to be no indication of the computational tools and software resources used to carry out the methods presented. These issues could be presented in a more orderly and clear manner)

Response: We are very sorry for ignoring related calculated tools and software resources and this paper has added these contents according to the Reviewer’s suggestion on line 304 in the paper and marked in red.

14. Response to comment: (Figure 8 does not seem to be very clear with regard to the characteristics)

Response: Considering the Reviewer’s suggestion, we have supplemented the relevant characteristics in Figure 8.

15. Response to comment: (After all that has been read, this technique can be considered as a complement to target tracking and displacement measurement, but it certainly does not seem to be a substitute for this work in its current state)

Response: We have made the correction “This DPG algorithm can be considered as a complement to target tracking and displacement measurement” according to the Reviewer’s suggestion on line 442 in the paper and marked in red.

16. Response to comment: (I recommend including the limitations regarding the consideration of the calculation speed and stability of the proposed matching algorithm.)

Response: As Reviewer suggested that we have supplemented the limitations “The proposed DPG method also has certain limitations, such as calculation speed and stability, under certain noise conditions” regarding the consideration of the calculation speed and stability according to the Reviewer’s suggestion on line 444 in the paper and marked in red.

Other changes:

1. Line 10, the statements of “a series images in the DIC approach” were corrected as “a series of images in the DIC approach”

2. Line 14, the statements of “the traditional gradient method is used to…” were corrected as “the traditional gradient-based method is used to…”

3. Line 16, the statements of “…direction are both utilized as interpolation center” were corrected as “…direction are both utilized as an interpolation center”

4. Line 44, the statements of “and performance of these methods been…” were corrected as “and the performance of these methods has been…”

5. Line 66, the statements of “The accuracy of GB method…” were corrected as “The accuracy of the GB method…”

6. Line 74, the statements of “whose accuracy are determined by center selection” were corrected as “whose accuracy is determined by center selection”.

7. Line 81, the statements of “then the identified displacement is linear combined” were corrected as “then the identified displacement is linearly combined”.

8. Line 83, the statements of “but also improves accuracy…” were corrected as “but also improves the accuracy…”.

9. Line 189, the statements of “lower bars represents the 90% deviation lines” were corrected as “lower bars represent the 90% deviation lines”.

10. Line 338, the statements of “height and wide of the beam is 1400,100 and 50 mm respectively” were corrected as “height and width of the beam are 1400,100 and 50 mm respectively”.

11. Line 347, the statements of “at 8-bit gray levels were record through…” were corrected as “at 8-bit gray levels recorded through…”.

12. Line 350, the statements of “camera after setting up the experimental environmental” were corrected as “camera after setting up the experimental environment”.

13. Line 358, the statements of “with red dot line near the micrometers is considered” were corrected as “with a red dot line near the micrometers is considered”.

14. Line 386, the statements of “displacements identified by GB method differs…” were corrected as “displacements identified by GB method differ…”

15. Line 389, the statements of “The DPG method not only achieve high accuracy…” were corrected as “The DPG method not only achieves high accuracy…”

16. Line 396, the statements of “beyond the scope this study” were corrected as “beyond the scope of this study”

17. Line 419, the statements of “indicating accuracy of the GB method is greatly depend on” were corrected as “indicating accuracy of the GB method is greatly dependent on”

18. Line 468-471, “Baqersad J, Poozesh P, Niezrecki C, et al. Photogrammetry and optical methods in structural dynamics–A review. Mechanical Systems and Signal Processing, 2016, S0888327016000388–1:18. doi:10.1016/j.ymssp.2016.02.011 

Feng D, Feng M Q. Computer vision for SHM of civil infrastructure: From dynamic response measurement to damage detection–A review. Engineering Structures, 2018, 156(FEB.1):105-117. doi:10.1016/j.engstruct.2017.11.018” was added  

19. Line 477-485, “Chen M, Tang Y C, Zou X, et al. High-accuracy multi-camera reconstruction enhanced by adaptive point cloud correction algorithm. Optics and Lasers in Engineering, 2019, 122:170-183. doi:10.1016/j.optlaseng.2019.06.011

Tang Y, Li L, Wang C, et al. Real-time detection of surface deformation and strain in recycled aggregate concrete-filled steel tubular columns via four-ocular vision. Robotics and Computer-Integrated Manufacturing, 2019, 59: 36-46. doi: 10.1016/j.rcim.2019.03.001

Tang Y, Chen M, Lin Y, et al. Vision-Based Three-Dimensional Reconstruction and Monitoring of Large-Scale Steel Tubular Structures. Advances in Civil Engineering, 2020, 1–17. (Prepublish) doi:10.1155/2020/1236021

Rizo-Patron S , Sirohi J . Operational Modal Analysis of a Helicopter Rotor Blade Using Digital Image Correlation. Experimental Mechanics, 2016, 57(3):1-9. doi:10.1007/s11340-016-0230-6” was added.  

20. Line 508-509, “BING P, HUIMIN X, ZHAO Y. Equivalence of digital image correlation criteria for pattern matching. Applied Optics, 2010,49(28):5501. doi:10.1364/AO.49.005501” was added.

21. Line 523-526, “Orujov F, Maskeliunas R, Damasevicius R, et al. Fuzzy based image edge detection algorithm for blood vessel detection in retinal images. Applied Soft Computing, 2020, 94:106452. doi: 10.1016/j.asoc.2020.106452

Versaci M, Morabito F C. Image Edge Detection: A New Approach Based on Fuzzy Entropy and Fuzzy Divergence. International Journal of Fuzzy Systems, 2021,1-19. (Prepublish) doi: 10.1007/s40815-020-01030-5” was added.

We tried our best to improve the manuscript and made some changes in the manuscript. These changes will not influence the content and framework of the paper. And here we did not list the changes but marked in red in revised paper.

We appreciate for Editors/Reviewers’ warm work earnestly and hope that the correction will meet with approval.

Once again, thank you very much for your comments and suggestions.

Reviewer 3 Report

  • Some typos are present in the text so they need to be removed.
  • Some formulas are certainly not original, so it would be desirable to associate a relevant bibliographic reference to each of them.
  • The origin of Definition 1 is not very clear. So please specify.
  • The proposed approach is certainly interesting and deserves to be taken into consideration for any future research developments. However what if the images are affected by uncertainty and / or inaccuracy? Surely one should opt for techniques that exploit fuzzy logic-based approaches that are well suited for the preprocessing of images affected by uncertainty and / or inaccuracies. Notwithstanding that these techniques are not used in this work (but they can certainly be taken into consideration for any future developments) I advise the authors to include in the text at least one sentence that highlights this possibility by adding the following relevant works in the bibliography:

         doi: 10.1007/s40815-020-01030-5

         doi: 10.1016/j.asoc.2020.106452

Author Response

Responds to the reviewer’s comments:

Reviewer 3:

1. Response to comment: (Some typos are present in the text so they need to be removed)

Response: We have carried out a detailed and rigorous proofread of the entire article.

2. Response to comment: (Some formulas are certainly not original, so it would be desirable to associate a relevant bibliographic reference to each of them.)

Response: Considering the Reviewer’s suggestion, we have supplemented the relevant bibliographic reference to some formulas.

3. Response to comment: (The origin of Definition 1 is not very clear. So please specify)

Response: We are very sorry for our incorrect writing in definition 1 and this paper has corrected the definition 1 and added detailed information on line 103 in the paper and marked in red.

4. Response to comment: (The proposed approach is certainly interesting and deserves to be taken into consideration for any future research developments. However, what if the images are affected by uncertainty and / or inaccuracy? Surely one should opt for techniques that exploit fuzzy logic-based approaches that are well suited for the preprocessing of images affected by uncertainty and / or inaccuracies. Notwithstanding that these techniques are not used in this work (but they can certainly be taken into consideration for any future developments) I advise the authors to include in the text at least one sentence that highlights this possibility by adding the following relevant works in the bibliography: doi: 10.1007/s40815-020-01030-5 doi: 10.1016/j.asoc.2020.106452)

Response: As Reviewer suggested that we have supplemented the question of uncertainly “When using algorithmic processing, directly using the captured images for processing may be affected by uncertainty and/or inaccuracy. At this time, you can choose to use techniques based on fuzzy logic methods [29,30]. These methods are very suitable for Pre-processing of images affected by performance and/or imprecision. Although this technology was not used in this study (but these technologies can be considered in any future development)” on line 363 in the paper and the two papers given in the references on line 523 in the paper and marked in red.

Other changes:

1. Line 10, the statements of “a series images in the DIC approach” were corrected as “a series of images in the DIC approach”

2. Line 14, the statements of “the traditional gradient method is used to…” were corrected as “the traditional gradient-based method is used to…”

3. Line 16, the statements of “…direction are both utilized as interpolation center” were corrected as “…direction are both utilized as an interpolation center”

4. Line 44, the statements of “and performance of these methods been…” were corrected as “and the performance of these methods has been…”

5. Line 66, the statements of “The accuracy of GB method…” were corrected as “The accuracy of the GB method…”

6. Line 74, the statements of “whose accuracy are determined by center selection” were corrected as “whose accuracy is determined by center selection”.

7. Line 81, the statements of “then the identified displacement is linear combined” were corrected as “then the identified displacement is linearly combined”.

8. Line 83, the statements of “but also improves accuracy…” were corrected as “but also improves the accuracy…”.

9. Line 189, the statements of “lower bars represents the 90% deviation lines” were corrected as “lower bars represent the 90% deviation lines”.

10. Line 338, the statements of “height and wide of the beam is 1400,100 and 50 mm respectively” were corrected as “height and width of the beam are 1400,100 and 50 mm respectively”.

11. Line 347, the statements of “at 8-bit gray levels were record through…” were corrected as “at 8-bit gray levels recorded through…”.

12. Line 350, the statements of “camera after setting up the experimental environmental” were corrected as “camera after setting up the experimental environment”.

13. Line 358, the statements of “with red dot line near the micrometers is considered” were corrected as “with a red dot line near the micrometers is considered”.

14. Line 386, the statements of “displacements identified by GB method differs…” were corrected as “displacements identified by GB method differ…”

15. Line 389, the statements of “The DPG method not only achieve high accuracy…” were corrected as “The DPG method not only achieves high accuracy…”

16. Line 396, the statements of “beyond the scope this study” were corrected as “beyond the scope of this study”

17. Line 419, the statements of “indicating accuracy of the GB method is greatly depend on” were corrected as “indicating accuracy of the GB method is greatly dependent on”

18. Line 468-471, “Baqersad J, Poozesh P, Niezrecki C, et al. Photogrammetry and optical methods in structural dynamics–A review. Mechanical Systems and Signal Processing, 2016, S0888327016000388–1:18. doi:10.1016/j.ymssp.2016.02.011 

Feng D, Feng M Q. Computer vision for SHM of civil infrastructure: From dynamic response measurement to damage detection–A review. Engineering Structures, 2018, 156(FEB.1):105-117. doi:10.1016/j.engstruct.2017.11.018” was added  

19. Line 477-485, “Chen M, Tang Y C, Zou X, et al. High-accuracy multi-camera reconstruction enhanced by adaptive point cloud correction algorithm. Optics and Lasers in Engineering, 2019, 122:170-183. doi:10.1016/j.optlaseng.2019.06.011

Tang Y, Li L, Wang C, et al. Real-time detection of surface deformation and strain in recycled aggregate concrete-filled steel tubular columns via four-ocular vision. Robotics and Computer-Integrated Manufacturing, 2019, 59: 36-46. doi: 10.1016/j.rcim.2019.03.001

Tang Y, Chen M, Lin Y, et al. Vision-Based Three-Dimensional Reconstruction and Monitoring of Large-Scale Steel Tubular Structures. Advances in Civil Engineering, 2020, 1–17. (Prepublish) doi:10.1155/2020/1236021

Rizo-Patron S , Sirohi J . Operational Modal Analysis of a Helicopter Rotor Blade Using Digital Image Correlation. Experimental Mechanics, 2016, 57(3):1-9. doi:10.1007/s11340-016-0230-6” was added.  

20. Line 508-509, “BING P, HUIMIN X, ZHAO Y. Equivalence of digital image correlation criteria for pattern matching. Applied Optics, 2010,49(28):5501. doi:10.1364/AO.49.005501” was added.

21. Line 523-526, “Orujov F, Maskeliunas R, Damasevicius R, et al. Fuzzy based image edge detection algorithm for blood vessel detection in retinal images. Applied Soft Computing, 2020, 94:106452. doi: 10.1016/j.asoc.2020.106452

Versaci M, Morabito F C. Image Edge Detection: A New Approach Based on Fuzzy Entropy and Fuzzy Divergence. International Journal of Fuzzy Systems, 2021,1-19. (Prepublish) doi: 10.1007/s40815-020-01030-5” was added.

We tried our best to improve the manuscript and made some changes in the manuscript. These changes will not influence the content and framework of the paper. And here we did not list the changes but marked in red in revised paper.

We appreciate for Editors/Reviewers’ warm work earnestly and hope that the correction will meet with approval.

Once again, thank you very much for your comments and suggestions.

Round 2

Reviewer 1 Report

The authors addressed my comments partly. Point 2 (related to the levels of noise) and point 4 from my previous report are still not clear. 

1. For point 2 the authors didn't address my previous comment about the levels of noise in images.

2. For point 4 (related to Subpart 4.2.3) the authors should present mathematical proofs or more experiments to support their conclusions. Now their conclusions are not relevant due to lack of these.

Author Response

1. Response to comment: (For point 2 the authors didn't address my previous comment about the levels of noise in images.)

Response: It is really true as Reviewer suggested that this paper adds the PSNR of noise in line 298 and Figure 8(b).

2. Response to comment: (For point 4 (related to Subpart 4.2.3) the authors should present mathematical proofs or more experiments to support their conclusions. Now their conclusions are not relevant due to lack of these.)

Response: It is really true as Reviewer suggested that this paper use mean error to evaluate the recognition result of the illumination analysis, as shown in Figure 15. Moreover, Figure 15 is explained in line 417.

Other changes:

1. Line 8, the word of “has” was corrected as “have”.

2. Line 16, the word of “displacements” was corrected as “displacement”.

3. Line 30, the word of “has” was corrected as “have”.

4. Line 66, the word of “its” was corrected as “the”.

5. Line 87, the word of “about” was corrected as “of”.

6. Line 120, the statements of “the pre-estimated of the displacement” were corrected as “the pre-estimated displacement”.

7. Line 141, the statements of “Taylor expansions of function g(•)” were corrected as “Taylor’s expansion of function g(•)”.

8. Line 149, the word of “expression” was corrected as “expressions”.

9. Line 191, the statements of “and interpolation center exceeded 0.5 pixel.” were corrected as “and the interpolation center exceeded 0.5 pixel.”.

10. Line 208, the statements of “to generate the other five image sets” were corrected as “to generate five other image sets”.

11. Line 224, the statements of “via weight coefficient to obtain the final identification results” were corrected as “via a weight coefficient to obtain the final identification results”.

12. Line 229, the statements of “using the left and right pixel as interpolation center respectively” were corrected as “using the left and right pixel as an interpolation center respectively”.

13. Line 266, the statements of “The detail procedure of the proposed DPG method is as followings,” were corrected as “The detailed procedure of the proposed DPG method is as follows,”.

14. Line 283, the word of “calculation” was corrected as “calculated”

15. Line 284, the statements of “it is not be described in detail here” were corrected as “it will not be described in detail here”

16. Line 291, the word of “is” was corrected as “are”.

17. Line 296, the statements of “dose not be use” were corrected as “dose is not be use”

18. Line 397, the word of “algorithm” was corrected as “algorithms”.

19.Line 436, the statements of “displacements are obtained from these two integral pixels” were corrected as “displacement is obtained from these two integral pixels”.

We tried our best to improve the manuscript and made some changes in the manuscript. These changes will not influence the content and framework of the paper. And here we did not list the changes but marked in red in revised paper.

We appreciate for Editors/Reviewers’ warm work earnestly and hope that the correction will meet with approval.

Once again, thank you very much for your comments and suggestions.

Reviewer 2 Report

accept

Author Response

1. Line 8, the word of “has” was corrected as “have”.

2. Line 16, the word of “displacements” was corrected as “displacement”.

3. Line 30, the word of “has” was corrected as “have”.

4. Line 66, the word of “its” was corrected as “the”.

5. Line 87, the word of “about” was corrected as “of”.

6. Line 120, the statements of “the pre-estimated of the displacement” were corrected as “the pre-estimated displacement”.

7. Line 141, the statements of “Taylor expansions of function g(•)” were corrected as “Taylor’s expansion of function g(•)”.

8. Line 149, the word of “expression” was corrected as “expressions”.

9. Line 191, the statements of “and interpolation center exceeded 0.5 pixel.” were corrected as “and the interpolation center exceeded 0.5 pixel.”.

10. Line 208, the statements of “to generate the other five image sets” were corrected as “to generate five other image sets”.

11. Line 224, the statements of “via weight coefficient to obtain the final identification results” were corrected as “via a weight coefficient to obtain the final identification results”.

12. Line 229, the statements of “using the left and right pixel as interpolation center respectively” were corrected as “using the left and right pixel as an interpolation center respectively”.

13. Line 266, the statements of “The detail procedure of the proposed DPG method is as followings,” were corrected as “The detailed procedure of the proposed DPG method is as follows,”.

14. Line 283, the word of “calculation” was corrected as “calculated”

15. Line 284, the statements of “it is not be described in detail here” were corrected as “it will not be described in detail here”

16. Line 291, the word of “is” was corrected as “are”.

17. Line 296, the statements of “dose not be use” were corrected as “dose is not be use”

18. Line 397, the word of “algorithm” was corrected as “algorithms”.

19. Line 436, the statements of “displacements are obtained from these two integral pixels” were corrected as “displacement is obtained from these two integral pixels”.

We tried our best to improve the manuscript and made some changes in the manuscript. These changes will not influence the content and framework of the paper. And here we did not list the changes but marked in red in revised paper.

We appreciate for Editors/Reviewers’ warm work earnestly and hope that the correction will meet with approval.

Once again, thank you very much for your comments and suggestions.

Round 3

Reviewer 1 Report

The authors addressed all my comments. The paper can be accepted in the present form.